# Phenotyping of Rare *CFTR* Mutations Reveals Distinct Trafficking and Functional Defects

**DOI:** 10.3390/cells9030754

**Published:** 2020-03-19

**Authors:** Marjolein Ensinck, Liesbeth De Keersmaecker, Lise Heylen, Anabela S. Ramalho, Rik Gijsbers, Ricard Farré, Kris De Boeck, Frauke Christ, Zeger Debyser, Marianne S. Carlon

**Affiliations:** 1Molecular Virology and Gene Therapy, Department of Pharmaceutical and Pharmacological Sciences, KU Leuven, 3000 Leuven, Flanders, Belgium; marjolein.ensinck@kuleuven.be (M.E.); liesbeth.dekeersmaecker@kuleuven.be (L.D.K.); lise.heylen@kuleuven.be (L.H.); rik.gijsbers@kuleuven.be (R.G.); frauke.christ@kuleuven.be (F.C.); zeger.debyser@kuleuven.be (Z.D.); 2CF Centre, Woman and Child Unit, Department of Development and Regeneration, KU Leuven, 3000 Leuven, Flanders, Belgium; anabela.santoramalhovenancio@kuleuven.be; 3Leuven Viral Vector Core, KU Leuven, 3000 Leuven, Flanders, Belgium; 4Translational Research in Gastrointestinal Disorders (TARGID), KU Leuven, 3000 Leuven, Flanders, Belgium; ricard.farre@kuleuven.be; 5University Hospital Leuven, 3000 Leuven, Flanders, Belgium; Christiane.DeBoeck@uzleuven.be

**Keywords:** rare *CFTR* mutations, CFTR maturation, CFTR function, CFTR trafficking, CFTR modulators

## Abstract

*Background.* The most common *CFTR* mutation, F508del, presents with multiple cellular defects. However, the possible multiple defects caused by many rarer *CFTR* mutations are not well studied. We investigated four rare *CFTR* mutations E60K, G85E, E92K and A455E against well-characterized mutations, F508del and G551D, and their responses to corrector VX-809 and/or potentiator VX-770. *Methods.* Using complementary assays in HEK293T stable cell lines, we determined maturation by Western blotting, trafficking by flow cytometry using extracellular 3HA-tagged CFTR, and function by halide-sensitive YFP quenching. In the forskolin-induced swelling assay in intestinal organoids, we validated the effect of tagged versus endogenous CFTR. *Results.* Treatment with VX-809 significantly restored maturation, PM localization and function of both E60K and E92K. Mechanistically, VX-809 not only raised the total amount of CFTR, but significantly increased the traffic efficiency, which was not the case for A455E. G85E was refractory to VX-809 and VX-770 treatment. *Conclusions.* Since no single model or assay allows deciphering all defects at once, we propose a combination of phenotypic assays to collect rapid and early insights into the multiple defects of CFTR variants.

## 1. Introduction

Cystic fibrosis (CF) is a hereditary, progressive multi-organ disorder affecting >85,000 people worldwide. Lung disease, however, is the major cause of morbidity and mortality. Despite better symptomatic care, the median age at death remains in the early thirties [1]. CF is caused by mutations in the Cystic Fibrosis Transmembrane Conductance Regulator (*CFTR*) gene [2,3,4], which encodes an anion channel expressed in the apical membrane of epithelial cells. More than 2000 mutations have been described (www.genet.sickkids.on.ca), of which at least 352 are disease-causing (www.cftr2.org). *CFTR* mutations disturb the synthesis, function or stability of the protein at the plasma membrane (PM). The most common mutation, F508del, results in a loss of phenylalanine at position 508 and accounts for 70% of CF alleles. By comparison, all other mutations are rare. *CFTR* mutations are grouped into classes according to how the mutation disturbs the production (class I, II, V, VII), function (class III, IV) or stability (class VI) of the CFTR protein [5]. Many mutations, however, cause multiple defects [6].

CFTR modulators have been developed to improve mutant CFTR folding and trafficking (“correctors”) or to potentiate channel opening at the PM (“potentiators”). Until recently, only three CFTR modulators were market approved for mono or combination therapy: potentiator ivacaftor (VX-770) and correctors lumacaftor (VX-809) and its derived molecule tezacaftor (VX-661). Together, they offer a treatment to approximately half of the CF patient population, with moderate to impressive clinical benefit, depending on the mutation present. With the recent FDA approval of Trikafta^™^, a triple combination therapy consisting of two distinct CFTR correctors, tezacaftor and elexacaftor (VX-445), and potentiator ivacaftor, 82 to 90% of CF patients will soon qualify for a CFTR-targeting therapy. Trikafta^™^ was shown to significantly improve lung function and overall clinical benefit in patients carrying one or two F508del alleles [7,8]. This leaves 10% of CF patients without causal treatment. These are patients with two minimal function alleles, such as nonsense mutations, insertion and deletion mutations, canonical splicing mutations, and processing mutations other than F508del. Due to their low prevalence, these rare mutations have not been thoroughly characterized. A proper characterization of their molecular defects is however indispensable for tailoring novel treatments to these patient groups.

In this study, we aimed at unraveling the molecular defects of four rare *CFTR* mutations, E60K (p.Glu60Lys), G85E (p.Gly85Glu), E92K (p.Glu92Lys), A455E (p.Ala455Glu), all with a prevalence below 1% (www.cftr2.org). Only for A455E, CFTR modulator treatment is currently approved [9,10]. As F508del leads to pleiotropic molecular defects, ranging from impaired processing, trafficking and gating to reduced stability once rescued to the PM [6], we sought to comprehensively study the possibly multiple defects of these four rare mutations. Primary cell models represent a close proxy to the pathophysiology of human CF disease [11,12,13]. However, the complexity of their individual genotypes makes studying the molecular consequences of a particular mutation challenging, especially for rare *CFTR* mutations which rarely present as homozygous. Interestingly, the FDA expanded its approval of Kalydeco® (ivacaftor monotherapy for CF patients with gating or residual function mutations) for additional mutations mainly based on in vitro data in Fisher Rat Thyroid (FRT) cells [9,14], underscoring the positive predictive value of cell lines for response to treatment [15].

Here, we provide a rapid and versatile strategy to pinpoint specific molecular defects in relatively rare, not fully characterized *CFTR* mutations, and investigate potential rescuing therapies by combining three complementary assays on maturation, trafficking and function in human cell lines. Our results not only identify novel potential responders to therapy (i.e., functional rescue of E60K and E92K by VX-809), but also provide mechanistic insights into how VX-809 rescues trafficking and hence function for the respective mutants studied. Ultimately, a combination of models and assays will bring us closer to faithfully unraveling the molecular defects of *CFTR* mutations and their responses to novel CFTR targeting drugs.

## 2. Materials and Methods

### 2.1. Cystic Fibrosis Transmembrane Conductance Regulator (CFTR) Lentiviral Vector (LV) Production

Starting from the pCHMWS-CFTR_full-length plasmid [16] into which a puromycin resistance gene was added linked to CFTR with an internal ribosomal entry site (IRES), the different CFTR variants were cloned using gBlock Gene Fragments (IDT, Haasrode, Belgium). All variants were also cloned to contain a triple hemagglutinin (3HA) tag in the fourth extracellular loop (EC-loop) of CFTR [17]. LV production was performed as described before [16].

### 2.2. Human Embryonic Kidney 293T (HEK293T) Stable Cell Lines 

#### 2.2.1. Stable Cell Line Generation

HEK293T (CRL-3216; ATCC, Manassas, VA, USA) cell lines stably expressing the different (tagged) CFTR variants were generated by lentiviral transduction, using a serial 1/3 dilution of vector. Three days post transduction puromycin selection (2 µg/mL) was started and cells were kept on puromycin after that. Cells transduced with the lowest amount of vector which survived puromycin selection were expanded and used in all experiments, assuming a single integration event in each cell.

#### 2.2.2. RT-PCR for CFTR mRNA Expression Levels

RNA was isolated using Aurum Total RNA mini kit (#7326820, Bio-Rad, Hercules, CA, USA) and cDNA was prepared with High Capacity cDNA Reverse Transcription Kit (#4368813, Thermo Fischer Scientific, Waltham, MA, USA). Each reaction contained LightCycler® 480 SYBR Green I Master (#4707516001, Roche, Basel, Switzerland), and primers for either CFTR (Fw: GCAGTTGATGTGCTTGGCTA; Rev: ACTGCCGCACTTTGTTCTCT) or β-actin (Fw: TCACCCACACTGTGCCCATCTACGA; Rev: CAGCGGAACCGCTCATTGCCAATGG). Samples were run in LightCycler 480 (Roche) and analyzed using its software. *CFTR* mRNA expression was normalized to that of β-actin, and put relative to 3HA-WT. For the quantification of Western blot, flow cytometry and halide sensitive YFP, data were normalized on *CFTR* mRNA expression levels.

#### 2.2.3. Western Blotting

A total of 15 µg total protein of crude cell extracts prepared in 1% SDS with complete protease inhibitors (#11873580001, Roche) was separated by SDS–PAGE in a 3–8% Tris-acetate gel (NuPAGE, #WG1602A, Invitrogen, Waltham, MA, USA) and transferred to a PVDF membrane. The blot was incubated with a mix of three mouse monoclonal CFTR antibodies (570, 596, 660) (CFFT, 1:1000 each, [18]). α-tubulin (#T5168, Sigma-Aldrich, St. Louis, MO, USA, 1:5000) was used as loading control. Proteins were visualized after HRP secondary antibody (#P0447, Dako, Glostrup, Denmark, 1:10.000) labelling using Clarity Western ECL substrate (#1705061, Bio-Rad) and imaged using the Fuijifilm LAS 3000 mini (Fujifilm, Tokyo, Japan). Western blots were quantified using ImageQuant TL (GE Healthcare, Chicago, IL, USA) software.

#### 2.2.4. Immunocytochemistry

For PM staining, cells were blocked with 1% BSA-PBS and incubated with HA.11 antibody (#901515, Biolegend, San Diego, CA, USA, 1:1000) at 4 °C on living cells. Next, cells were fixed (4% PFA) followed by Alexa-488 secondary antibody (#A-11001, Thermo Fischer Scientific, 1:500). Next, cells were fixed again, followed by permeabilization and HA.11 primary antibody (1:1000). Total anti-HA stained CFTR was visualized using Alexa-633 secondary antibody (#A-21050, Thermo Fischer Scientific, 1:500). Nuclei were stained with DAPI (4′,6-diamidino-2-fenylindole, #D1306, Thermo Fischer Scientific, 1:2000) and sections analyzed by confocal microscopy. 

#### 2.2.5. Flow Cytometry

Cells were seeded in 96-well plates. VX-809 (#S1565, Selleckchem, Houston, TX, USA, 2.5 µM) or dimethylsulfoxide (DMSO) was added 24 h later for selected experiments. For the dose-response of VX-809, cells were treated with a concentration range of 50 nM to 50 µM with constant DMSO. 24 h later, cells were dissociated to single cells with 0.25% Trypsin-EDTA for 2 min at 37 °C and transferred to a conical 96-well plate. PM and total staining were performed in parallel. For PM staining, cells were stained with HA.11 antibody (#901515, Biolegend 1:1000) at 4 °C, then fixed with IC fixation buffer (#00-8222-49, eBioscience, Waltham, MA, USA) and followed by Alexa-488 secondary antibody (#A-11001, Thermo Fischer Scientific, 1:500). For total staining cells were fixed and permeabilized prior to primary antibody. Per well 10.000 events were measured with the Guava easyCyte HT and analyzed in Guava InCyte. HEK293T cells, not overexpressing any CFTR variant were used as a negative control to determine the positive fraction. Plasma membrane density was determined as the percentage of PM CFTR positive cells (%gated) * Median Fluorescent Intensity of the PM CFTR (MFI), relative to WT CFTR. Traffic efficiency (TE) by VX-809 was compared to DMSO (ΔTE = TE_VX-809_ − TE_DMSO_), with TE = % PM/ % Total CFTR (% of WT DMSO).

#### 2.2.6. Halide Sensitive (HS) YFP Assay

The assay was performed as detailed in [19]. Briefly, cells were transfected with a plasmid encoding a halide sensitive yellow fluorescent protein (YFP-H148Q/I152L/F46L) [20] using PEI (polyethylenimine) and immediately plated into black, clear-bottomed 96-well plates coated with Poly-D-Lysine. After overnight incubation VX-809 (2.5 µM) or DMSO was added for another 24 h. Next, the cells were washed with DPBS and potentiator VX-770 (#S1144, Selleckchem, 3 µM) and/or CFTR activator forskolin (#F3917, Sigma-Aldrich, 10 µM) was added for 20 min. Fluorescence was measured using Perkin Elmer Envision for 5 sec after which a I^−^ buffer (137 mM NaI, 2.7 mM KI, 1.7 mM KH_2_PO_4_, 10.1 mM Na_2_HPO_4_, 5 mM d-glucose) was injected into the well and fluorescence monitored for another 7 sec. YFP quenching was determined at the end of the interval as F/F_0_, and CFTR function as 1−(F/F_0_).

### 2.3. Intestinal Organoids

#### 2.3.1. Viral Vector Transduction

Human intestinal organoids were trypsinized to single cells, resuspended with equal amounts of viral vector and Matrigel (#356231, Corning, Corning, NY, USA) and grown in complete organoid medium [12] containing 10 µM Rock inhibitor (Y-27632-2HCl, #Y0503, Sigma-Aldrich) for the first three days. Ethical approval is in place for collection of rectal mucosa tissue for generating intestinal organoids (Ethics Review Board of the University Hospitals Leuven—S56329, approved on July 16th, 2018). All patients/parents gave written informed consent and/or assent.

#### 2.3.2. Forskolin Induced Swelling (FIS) Assay to Quantify CFTR Activity

14 days post-transduction, FIS was performed as described previously [12,16] with minor modifications. VX-809 (2.5 µM) or DMSO was added to specific wells 24 h before FIS. Organoids were stimulated with forskolin (5 µM) and VX-770 (3 µM) or DMSO, and analyzed by confocal live cell microscopy at 37 °C for 60 min (LSM800, Zeiss, Jena, Germany, Zen Blue). The total organoid area increase relative to *t* = 0 of forskolin treatment was quantified.

### 2.4. Statistics

All data were presented as mean ± S.D. either for a single, representative experiment or for independent, replicate experiments. Pearson’s correlations and the EC_50_ of the dose-response of VX-809 on 3HA-F508del-CFTR were calculated in Graphpad Prism 8.2. Software Inc. (San Diego, CA, USA). The effect of compound treatment over DMSO was compared by multiple t-tests for each CFTR mutant in grouped data sets (DMSO vs. treatment), a *P*-value < 0.05 was considered statistically significant.

## 3. Results

### 3.1. E60K, G85E, E92K and A455E Lead to Impaired Processing and Endoplasmic Reticulum Retention of CFTR

To study the pleiotropic molecular consequences of four rare *CFTR* mutations, E60K, G85E, E92K and A455E (Figure 1A), we resorted to human HEK293T cells stably overexpressing CFTR after lentiviral vector (LV) transduction, rather than patient-derived cell models, as this allows to study mutations in the absence of a confounding mutation on the other allele. Furthermore, HEK293T cells do not express any CFTR protein (Figure 1B,C and Appendix A—first lane, ‘Neg’), allowing clean assessment of introduced CFTR variants.

To get a first insight into the biosynthetic pathway of these rare mutations, we determined CFTR processing and glycosylation indirectly by quantifying the amount of ER-located, immature, core glycosylated CFTR (band B) versus the level of fully mature, complex glycosylated post-Golgi CFTR (band C) via Western blots. As a reference, we took along gating mutant G551D, combined processing and gating mutant F508del and WT-CFTR. Common for all four rare CFTR mutants was the strongly reduced to absent ‘band C’, similar to F508del, and this in contrast to G551D and WT-CFTR (Figure 1B,D), underscoring a common maturation defect. Treatment with corrector VX-809 significantly promoted ER-exit and restored the exocytic pathway for E60K, E92K and, as well documented, for F508del [22,23] (Figure 1B,D). For A455E a smaller but significant rescue was observed, which was totally absent for G85E. Of note, quantification of CFTR maturation (and subsequent analyses in following figures) on HEK293T stable cell lines was normalized for each mutant to its respective *CFTR* mRNA level (RT-PCR data in Appendix A). Although Western blotting hereby provides valuable first insights into the biogenesis and maturation process of CFTR variants, the technique is inherently low-throughput and only indirectly addresses CFTR trafficking to the plasma membrane (PM) where it acts as a cAMP-regulated anion channel.

### 3.2. Tagging CFTR in the 4th Extracellular Loop Minimally Affects CFTR Maturation and Function

We therefore set out to accurately quantify CFTR trafficking by flow cytometry. As currently no antibodies allow detecting PM localized CFTR by recognition of extracellular CFTR epitopes, we resorted to introducing a 3HA epitope tag into the fourth extracellular loop as described previously [17]. To assess however that the tag does not interfere with CFTR maturation or function, we repeated the Western blot analysis for the tagged CFTR variants (Figure 1C and Appendix A). Although a slight reduction was observed in the amount of band C, most notably for A455E and F508del, quantification of CFTR processing in tagged variants correlated well with untagged variants both at baseline and after VX-809 correction (R^2^~0.9, *P* = 0.002; Figure 1E and Appendix A). Also for CFTR function, measured as the degree of halide sensitive YFP quenching (HS-YFP), tagged and untagged CFTR variants correlated nicely at baseline and after VX-809 corrector, VX-770 potentiator or combination treatment (R^2^~0.9, *P* < 0.01, Figure 2A–C and Appendix A). As an additional control, we assessed the effect of tagging CFTR in an extracellular loop on CFTR function in patient-derived intestinal organoids, exemplified on F508del, as this allows a direct comparison to endogenous F508del, for which CFTR modulator responses are well documented [22,23]. To do so, we transduced CF organoids (genotype N1303K/3121-1G > A), which are non-responsive to VX-809 and/or VX-770 (Appendix A), with an LV to stably overexpress 3HA-F508del and assessed functional responses (i.e., CFTR-dependent organoid swelling [12]) to above mentioned modulators. Comparing F508del/F508del organoids (Figure 2D; ‘Endo’: top row, E: open circles) to N1303K/3121-1G > A organoids overexpressing 3HA-F508del (Figure 2D: ‘3HA-’: bottom row, E: closed circles), CFTR functional responses were near-identical. In conclusion, with this comprehensive analysis of a possible tag effect on CFTR, we reassuringly show that CFTR maturation and function are minimally affected for the different mutations studied, validating its use for complementary trafficking studies.

### 3.3. Corrector VX-809 Restores E60K and E92K Function

Besides allowing to correlate functional responses for tagged and untagged CFTR variants, HS-YFP quenching identified potential responders to CFTR modulator therapy (Figure 3A–C). In particular, combined VX-809 and VX-770 treatment (market approved as Orkambi®) significantly increased CFTR function of both E60K and E92K, reaching ~80% of WT-CFTR (*P* < 0.0001, Figure 3C), similar to the potentiator rescue of VX-770 (Kalydeco®) on gating mutant G551D (*P* < 0.0001, Figure 3B). In line with VX-809’s ability to correct CFTR maturation of both mutants (Figure 1B–D), VX-809 was the sole main contributor to a functional rescue (Figure 3A), mechanistically suggesting restored CFTR folding in the ER and subsequently, a full recovery of CFTR activity once rescued to the PM. This is in contrast to F508del and A455E, where combined VX-809 and VX-770 provided an additive benefit to CFTR function. G85E function could not be restored in any condition tested.

### 3.4. A Medium-Throughput Flow Cytometry Assay for Quantifying CFTR Plasma Membrane Density and Traffic Efficiency

Flow cytometry has the potential to provide rapid and quantitative information on cellular processes of thousands of individual cells by fluorescent labeling. While extracellular CFTR tags have already been used in the context of quantifying global CFTR PM densities by microscopy, ELISA or flow cytometry [24,25,26,27], we aimed at detailing the sensitivity of the flow cytometry-based trafficking assay by including reference compounds on well-characterized *CFTR* mutations and expanding its use to mechanistic insights into traffic efficiency. First, we confirmed 3HA-CFTR expression and localization for all CFTR mutants by confocal microscopy, sequentially visualizing first PM (in green), followed by total CFTR (in red) (Figure 4A,B). This visually depicted an increase in PM density after VX-809 treatment for mutants E60K, E92K, F508del and to lesser extent A455E, although for none to the level of WT-CFTR expression (Figure 4B). Next, we set out to accurately quantify these differences by flow cytometry. We started by determining the dose-response corrective effect of VX-809 on F508del trafficking to the PM to properly validate the assay. Indeed, VX-809 treatment led to a dose-dependent increase in F508del PM density with a corresponding EC_50_ of 345 nM (Figure 5A and Appendix A), similar to that obtained in maturation studies [23]. Next, we quantified the PM density of the four rare CFTR mutants studied before and after VX-809 treatment, compared to WT-CFTR and gating mutant G551D, for which processing and trafficking should be unaffected (Figure 5B–E). VX-809 is known to stabilize the nascent CFTR polypeptide chain during its co-translational folding in the ER and this in a mutation-unspecific manner [22], explaining the small increase in PM density also for WT and G551D (Figure 5C,E). Impressively though, VX-809 restored E60K and E92K trafficking significantly by 27- and 56-fold, respectively (Figure 5C,E), in line with its rescue in CFTR maturation and function (Figure 1B–D and Figure 3A). Relative to WT-CFTR, all four CFTR mutants showed minimal residual PM CFTR (from 0.2% for G85E to 3.9% for F508del) (Figure 5C; grey bars/circles) and strongly decreased total CFTR levels (from 3.5% for G85E and E92K to 8.0% for F508del) (Figure 5D; grey bars/circles), indicative of a pronounced processing defect inducing rapid degradation of misfolded protein. To answer the question whether VX-809 acts through prevention of degradation (which would mainly increase total CFTR), or by promoting ER-exit and trafficking to the PM (increase in PM density > increase in total CFTR), we calculated the traffic efficiency for the respective CFTR mutants following VX-809 treatment (Figure 5F), which considers not only changes in PM but also in total CFTR. VX-809 significantly increased the traffic efficiency of both E60K and E92K compared to WT-CFTR (*P* < 0.01, ~4-fold for both), confirming mechanistically what was anticipated from its observed effect on CFTR maturation and function earlier (Figure 1B–D and Figure 3A). In line with published reports on F508del [22,23], VX-809 increased PM density and traffic efficiency modestly by 4- and 2-fold (Figure 5E,F), respectively, requiring an additional potentiator to restore its function (Figure 3B,C). VX-809 provided minor or no benefit to A455E and G85E traffic efficiency, respectively (Figure 5F). In conclusion, we here provide a thoroughly validated trafficking assay by flow cytometry, allowing medium-throughput (96-well) quantification of CFTR PM density and traffic efficiency. The assay thus offers early insights into specific molecular defects caused by *CFTR* mutations as well as the mode of action of CFTR modulating drugs. Additionally, the trafficking assay allows its implementation in drug discovery pipelines for identifying drug responses or drug targets.

## 4. Discussion

For the last 10% of CF patients without current treatment options provided by health authorities, there is an urgent need to keep investing in developing novel therapies, despite the low prevalence of each of these rare mutations. As it is well accepted that many *CFTR* mutations show pleiotropic defects [6], and that genotype-to-phenotype predictions in CF are challenging [28], complementary phenotypic assays still provide an easy way to gather early insights into the defects of CFTR variants. In this study, we combine three phenotypic assays to determine maturation, trafficking and function of four rare *CFTR* mutations, E60K, G85E, E92K and A455E. We build further on existing assays and in particular fully validate a flow cytometry based trafficking assay to gather rapid, quantitative and medium-throughput information on PM density and traffic efficiency of CFTR variants and their response to therapy. We identified two mutations, E60K and E92K, that might benefit from the approved combination treatment of corrector VX-809 and potentiator VX-770 (Orkambi®), and by extrapolation also Symdeko® and Trikafta^™^. Furthermore, we mechanistically showed that the significant functional rescue for both mutants originated mainly from restored trafficking and implied corrected folding, rather than from mere prevention of degradation by the ERAD. Our workflow of complementary assays therefore allows generating rapid and early insights into the defects and therapeutic responses of CFTR variants, directing research to further confirmatory studies in primary cell models that contain the complexity of CF patients’ genotypes.

The absence of mature CFTR (band C) for E92K and its significant functional rescue by VX-809 (Figure 1B–D), classify this mutant as a predominant class II processing mutant, in line with other reports obtained in human cell lines [22,29]. For the rare mutation E60K we to our knowledge provide a first report on a similar functional rescue as observed for E92K. While VX-809 monotherapy is not commercially available, its combination with potentiator VX-770 (Orkambi®) is, which from our data provided even stronger functional rescue, suggesting its clinical benefit in patients. However, chronic exposure to VX-770 was shown to downregulate VX-809 rescued PM expression of E92K in human CFBE cell lines [29], warranting caution on the added value of Orkambi® over lumacaftor monotherapy. Nevertheless, a strong rescue in CFTR function by VX-809 alone, and in combination with VX-770, was confirmed in patient-derived intestinal organoids harboring the E60K or E92K mutation [30], strengthening the predictive value of human cell lines for early therapeutic testing [15]. In contrast, G85E, which is located in the same transmembrane span 1 (TM1) as E92K, shows no residual PM CFTR and minimal rescue by any CFTR modulators to date (Figure 1, Figure 2, Figure 3, Figure 4 and Figure 5; [31,32]). It is clear that this mutation seems refractory to any available CFTR modulator drug, underscoring the need for alternative treatments for these patients, such as mutation-agnostic gene/mRNA therapies, modulation of alternative ion channels or small molecule induced ion channel strategies [16,33,34,35,36]. A455E is associated with milder CF than the other rare *CFTR* mutations studied and has multiple proposed defects, ranging from less protein (class V), to impaired processing (class II) and conductance (class IV) [5,6]. Our data similarly show strongly reduced complex glycosylated CFTR (band C; Figure 1B), PM and total CFTR (Figure 5B–D) at steady state conditions, underscoring rapid degradation of mutant protein by the ER and/or peripheral quality control. Cebotaru and colleagues identified the proteasome as the main source of protein degradation, indeed pointing towards a class II processing defect [37,38]. Both Kalydeco® (ivacaftor) and Symdeko® (tezacaftor, ivacaftor) are approved for patients with the A455E mutation (www.fda.gov, [9,10,14]). Our data further validate this decision, showing an additive effect of combined VX-809 and VX-770. Of note, VX-661 (tezacaftor) is derived from corrector VX-809, and was developed to reduce drug-drug interactions with VX-770 [39]. Its mode of action, however, is similar. Interestingly, VX-809 did not increase CFTR function by restoring traffic efficiency (Figure 5F), but rather by a suggested reduction in degradation, this in contrast to E60K and E92K. Besides their therapeutic potential, CFTR modulators thus allow insights into the defects of CFTR mutants, allowing in this example to differentiate VX-809’s mechanism of rescue for the different mutants studied.

While the maturation and indirectly the localization of the CFTR protein are classically determined via Western blotting, this technique is inherently labor intensive, low-throughput and semi-quantitative. An advantage is the possibility to determine CFTR biogenesis in primary cell samples. Extracellular tagged CFTR variants on the other hand allow direct visualization of its localization at the PM and quantification of PM density and traffic efficiency by ELISA, high content imaging or flow cytometry [24,25,26,27], but the tag might interfere with CFTR folding, trafficking or function. In this study, we therefore performed a thorough side by side comparison of seven untagged and tagged CFTR variants to address this possible concern. Values for CFTR maturation correlated significantly (Figure 1E), although some variability was observed. This could be due to the semi-quantitative nature of Western blotting or point towards a minor interference of the tag. In particular for A455E and F508del, band C (complex glycosylated fraction) appeared more diffuse for tagged compared to untagged CFTR, both at baseline and after VX-809 correction. Future in depth studies on the maturation of these mutants might provide additional insights into their complex glycosylation in the Golgi and subsequent stability at the PM and further elucidate possible tagging effects. Reassuringly however, functional measurements in both cell lines and intestinal organoids showed a strong correlation between tagged and untagged CFTR variants (Figure 2), supporting the use of extracellular 3HA-tagged CFTR for complementary trafficking studies. In that regard, we extensively validated a flow cytometry-based trafficking assay to gather rapid, sensitive, high-throughput and highly quantitative data on CFTR trafficking. Besides confirming the VX-809-mediated functional recovery of mutants E60K and E92K by quantifying a significant increase in PM density (Figure 5B–E), it ascribed this rescue predominantly to increased trafficking, rather than a mere prevention of protein degradation (Figure 5F). In our study, traffic efficiency was calculated from parallel PM and total CFTR staining. Alternatively, double-tagged CFTR constructs can be used, allowing easier upscaling for screening purposes as nicely exemplified in [25]. However, tagging CFTR should be verified on a case by case basis to confirm absence of adverse effects on CFTR biogenesis and function.

## 5. Conclusions

In conclusion, our study provides a workflow of complementary assays in stable cell lines for gaining rapid and early insights into the maturation, trafficking and functional defects of CFTR variants and their responses to novel treatments. In this way, research can be narrowed down towards further confirmatory studies in primary cell models which, on the one hand, contain the complexity of CF patients’ genotypes, but where phenotypic assays often are more challenging and throughput is limited. As there is currently no model system that fits all, both cell lines and primary cells still complement each other for providing personalized medicine to all CF patients in the future.

## Figures and Tables

**Figure 1 cells-09-00754-f001:**
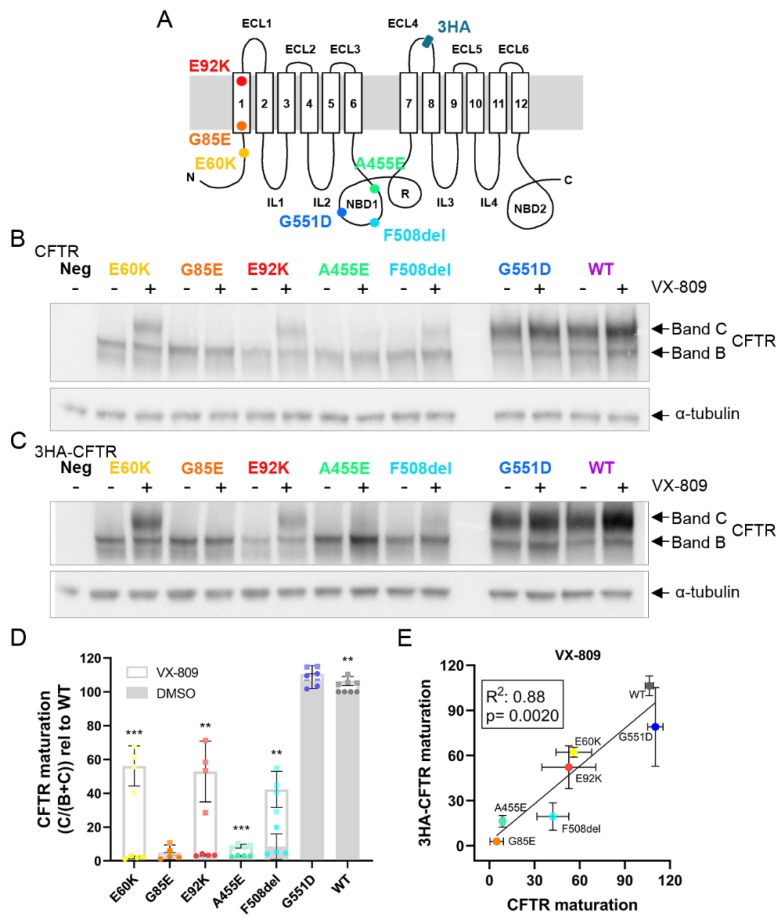
Assessment of CFTR maturation of rare CFTR mutants and their response to corrector VX-809. (**A**) Schematic CFTR protein structure adapted from reference [21], depicting the (rare) CFTR mutants studied and the localization of the 3HA tag in the fourth extracellular loop [17]. Western blot analysis of (**B**) untagged and (**C**) 3HA-tagged CFTR variants by anti-CFTR antibodies and α-tubulin (loading control) in whole-cell lysates of stably transduced HEK293T cells. The lower band (‘band B’) represents immature, core-glycosylated CFTR, the upper band (‘band C’) the complex glycosylated, fully mature protein. (**D**) Rescue in CFTR maturation (without tag) by corrector VX-809 (2.5 µM, 24 h) expressed as the increase in band C over total (B + C) CFTR relative to WT. Dots (DMSO) and squares (VX-809) represent independent experiments, one of which is shown in (B). (**E**) Pearson correlation of the rescue in CFTR maturation by corrector VX-809 of untagged and 3HA-tagged CFTR variants, as quantified in (D). Graphs depict the mean ± S.D. of three (3HA−tagged) or four (untagged) independent experiments. VX-809 rescue over DMSO was compared for each mutant by multiple t-testing. ** *P* < 0.01, *** *P* < 0.001. ER: endoplasmic reticulum, 3HA: triple hemagglutinin, ECL: extracellular loop, IL: intracellular loop, NBD: nucleotide binding domain, MSD: membrane spanning domain, R: regulatory domain, LV: lentiviral vector.

**Figure 2 cells-09-00754-f002:**
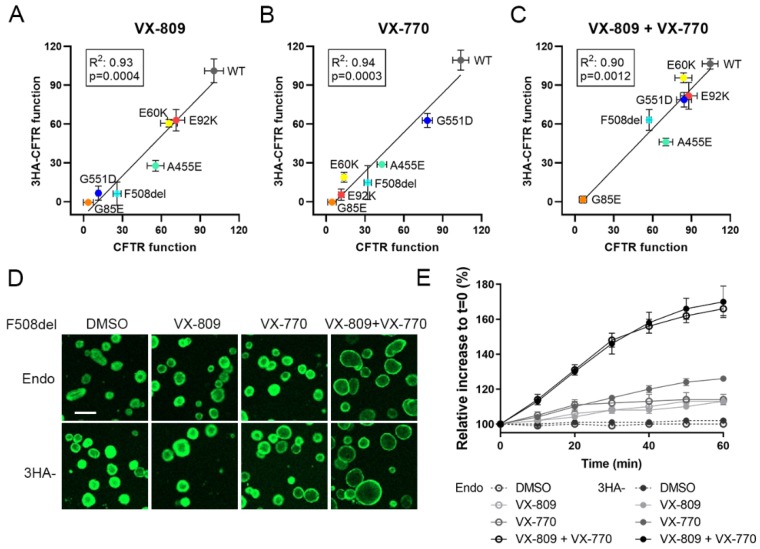
Study of the effect of tagging CFTR variants on CFTR function. (**A**–**C**) Pearson correlation of the rescue in CFTR function by corrector VX-809, potentiator VX-770 or the combination of both, of untagged and 3HA-tagged CFTR variants (for individual quantification, see Figure 3). HEK293T cells stably overexpressing CFTR variants were treated with 2.5 µM VX-809 for 24 h and/or 3 µM VX-770 for 20 min prior to determining CFTR function by HS-YFP quenching after stimulation with 10 µM forskolin (Fsk). Graphs depict the mean ± S.D. of three independent experiments. (**D**,**E**) Comparison of FIS responses of treated organoids with VX-809 (24 h, 2.5 µM), VX-770 (3 µM), VX-809+VX-770 or DMSO, all together with Fsk 5 µM. (**D**) Representative images of a FIS response for F508del/F508del CF organoids (top row, ‘Endo’; expressing endogenous F508del) and for CF organoids (genotype N1303K/3121-1G > A) stably overexpressing 3HA-F508del by LV transduction (bottom row, ‘3HA-‘), treated with DMSO, VX-809, VX-770 or both. Images represent *t* = 60 min after Fsk addition. Live organoids were visualized with calcein green. Scale bar = 200 µm. (**E**) Time course of both conditions over 60 min and responses to above CFTR modulators. A representative experiment is shown (n = 2 independent repeats). Mean ± S.D. of 4 wells per condition (~300 organoids/well). HS-YFP: halide sensitive yellow fluorescent protein, FIS: forskolin induced swelling, LV: lentiviral vector.

**Figure 3 cells-09-00754-f003:**
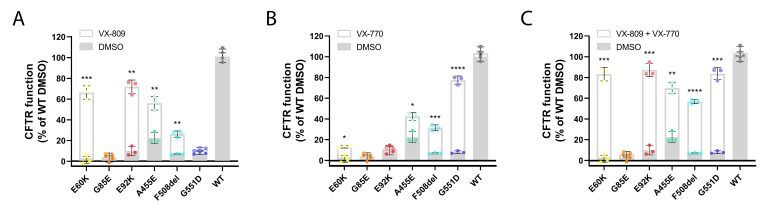
Rescue in CFTR function of rare CFTR mutants by VX-809 and/or VX-770. (**A**–**C**) HEK293T cells stably overexpressing untagged CFTR variants were treated with 2.5 µM VX-809 for 24 h (**A**), 3 µM VX-770 for 20 min (**B**) or a combination of both (**C**) prior to determining CFTR function by HS-YFP quenching after stimulation with 10 µM forskolin (Fsk). Graphs depict the mean ± S.D. of three independent experiments represented as dots (DMSO) or squares (compounds). Response to treatment over DMSO was compared for each mutant by multiple t-testing. * *P* < 0.05, ** *P* < 0.01, *** *P* < 0.001, **** *P* < 0.0001.

**Figure 4 cells-09-00754-f004:**
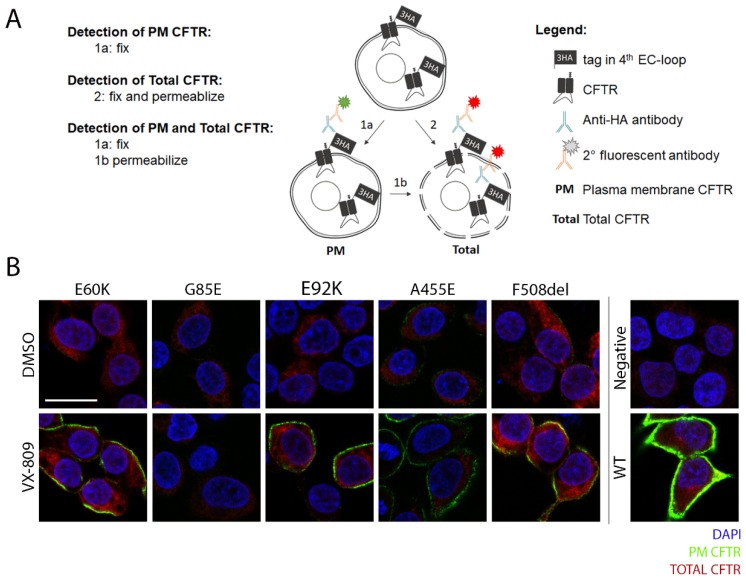
Mutant CFTR expression and subcellular localization. (**A**) Schematic overview of the staining procedure for detection of PM localized and/or total CFTR by its 3HA-tag. (**B**) Detection of PM localized (green) and total (red) 3HA-CFTR in stable HEK293T cell lines, visualized by sequential PM and total HA.11-antibody staining using confocal microscopy. Nuclei (blue) were stained with DAPI. Negative cells (VX-809 treated) and WT-CFTR (DMSO treated) are shown as a reference. Scale bar = 20 µm. 3HA: triple hemagglutinin, EC: extracellular, PM: plasma membrane.

**Figure 5 cells-09-00754-f005:**
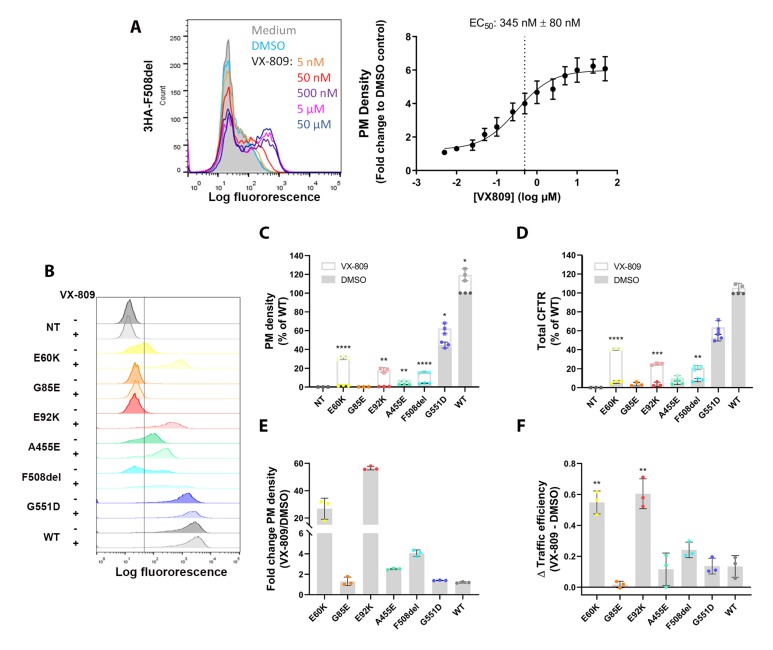
Quantification of plasma membrane density and traffic efficiency by flow cytometry of CFTR mutants. (**A**) 3HA-F508del-CFTR HEK293T cells were treated with increasing amounts of corrector VX-809 (range 5 nM–50 µM) or DMSO for 24 h and analyzed by flow cytometry. Left: Representative histograms of different VX-809 concentrations and their corresponding PM density levels is given. Right: EC_50_ determination of the dose-response to VX-809. Mean ± S.D. of n = 3 independent experiments. (**B**) Representative histograms of the PM density of the different 3HA-tagged CFTR variants under basal (DMSO) conditions or treated with VX-809. (**C** and **D**) PM density (**C**) and total CFTR (**D**) at baseline (DMSO) and after VX-809 treatment (2.5 µM, 24 h) relative to WT-CFTR DMSO. Independent experiments are shown as dots (DMSO) or squares (VX-809). (**E**) Fold increase in PM density after VX-809 treatment over DMSO control. (**F**) This graph represents the increase in traffic efficiency (TE) by VX-809 compared to DMSO (ΔTE = TE_VX-809_ − TE_DMSO_), with TE = % PM/ % Total CFTR (with WT DMSO = 1). Graphs depict the mean ± S.D. of three independent experiments. VX-809 rescue over DMSO was compared for each mutant by multiple t-testing. * *P* < 0.05, ** *P* < 0.01, *** *P* < 0.001, **** *P* < 0.0001. PM: plasma membrane, EC_50_: half-maximal effective concentration, NT: non-transduced.

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
