# Peer review of "Phenotyping of Rare CFTR Mutations Reveals Distinct Trafficking and Functional Defects"

_cells, 2020, doi:10.3390/cells9030754_

Round 1
Reviewer 1 Report
This paper is a nice example of characterising rare CF variants with respect to both biochemical/cellular properties, and responses to drugs.
The paper reports some interesting findings that are worthy of publication, to both general reader and CF researchers
I have a number of minor points which should be addressed prior to decision on acceptance.
L45, 64, 173, 336 (and others) - authors interchange rare and ultra rare without defining these paramaters. CFTR2 database (which they use) accurately lists allele frequency, so these values should be shown in paper and then a simple definition or rare and ultrarare stated, and used CONSISTENTLY throughout the paper.
Section 2.2 on cell lines should state whether or not cells were verified as having a single integration event, or merely highly likely to have a single integration event. FISH or PCR could be used to verify, but not clear if any such analysis was performed.
Figure 1A - is very nice, wonder if NBD1, R and NBD2 would look better if moved into the loops created in the intracellular regions. Suggestion, not a requirement.
Figure 1B and 1C would be better labelled with band C (golgi) and band B (ER). The CF field (and most of the text in the paper) refers to band B and band C (and band C/B ratio), so quite odd to label figure with just golgi and ER. Not even sure if golgi is correct, as I think band C refers to golgi and PM expression - happy to be corrected on this point by authors.
Figure 1D - graph of data from 1B and 1C is helpful, but I think this is data from only one of them, and not clear which. Is there a second set of data to be shown in the 1D. ALso, not clear what dots in the figure are meant to represent - most likely individual data points, but not clearly explained in figure legend.
Figure 2D needs minor attention to labels. “Combi” should be replaced with VX-809+VX-770. Also, nomenclature of the two rows needs some adjustment. Not clear that both rows are F508del. I think top row is F508del in an otherwise normal CFTR sequence, and bottom row is F508del in CFTR with the 3HA tag in EC loop4. THus, I think Endo means endogenous (but strictly speaking it’s not as its expressed from LV). It could be misinterpreted by some as endo(cytosis). I wonder if better term could be used.
F508del-3HA and F50del-no tag is one option?
Figure 3 - why the double X-axis on all three graphs?
Legend or labelling of 4B needs minor adjustment. Figure looks as though everything on top row is DMSO only so last column should express as it’s WT (or thats what you expect). HOwever, upper panel on far right is probably neg control (needs to state exactly what that is) and lower panel on far right is WT. HOwever, the use of neg/WT isn’t very clear!
Legend should also state cell type just because paper switches between several models so easier for reader when comparing data in figures quickly.
L389 Band C (golgi) is better than golgi (band C).
Only scientific concern, which authors may wish to comment on, regards the integration sites in the four different cell lines, and if this in anyway affects the functional data. I know functional assays are compared to mRNA levels which largely addresses this point. I wonder why they didn’t go for the FLP in system used by other CF groups? Was use of LV approach to allow direct comparison in organoids and cell lines?
It’s nice to see a cohort of data indicating that 3HA tag has no significant effect in context of several different variants, further validating this tag as a way to get round challenges faced by many in CF field of poor surface expression CFTR Abs.
Nice paper overall with valuable insights as to how CF community may find treatments for the rare (and ultra-rare?) variants.
Author Response
"Please see the attachment."

Reviewer 2 Report
The authors should be commended on submitting an excellent and well planned manuscript. The methodologies and strategies used to interrogate the research questions are sound and valid. However, there are minor issues which would need to be addressed prior the manuscript can be accepted for publication. The minor points are as listed below:
1) Line 135 stating "Cells not expressing CFTR...". The authors should expand on what these cells are
2) Line 142: "...using PEI and immediately...". The authors need to elaborate on what PEI is.
3) Line 143: "After ON incubation...". Expand on what ON means.
4) Figures 1D, 3, 5: Further details would need to be provided within the figure legends on what the data points (ie squares, circles) indicate.
5) Line 220: Figure citation needs to be kept consistent, ie Figure 2A-C before Figure S4B
Author Response
RESPONSE TO THE REVIEWERS’ COMMENTS:
REVIEWER 2.
C1. Line 135 stating "Cells not expressing CFTR...". The authors should expand on what these cells are
R1. This section was adapted to specify that these are HEK293T not overexpressing any CFTR variant.
HEK293T cells, not overexpressing any CFTR variant were used as a negative control to determine the positive fraction.
C2. Line 142: "...using PEI and immediately...". The authors need to elaborate on what PEI is
C3. Line 143: "After ON incubation...". Expand on what ON means.
R2+3. Section 2.2.6 was adapted in response to the reviewer’s comments (changes shown in red).
The assay was performed as detailed in [19]. Briefly, cells were transfected with a plasmid encoding a halide sensitive yellow fluorescent protein (YFP-H148Q/I152L/F46L) [20] using PEI (polyethylenimine) and immediately plated into black, clear-bottomed 96-well plates coated with Poly-D-Lysine. After overnight incubation VX-809 (2.5 µM) or DMSO was added for another 24 h. Next, the cells were washed with DPBS and potentiator VX-770 (#S1144, Selleckchem, 3 µM) and/or CFTR activator forskolin (#F3917, Sigma-Aldrich, 10 µM) was added for 20 min. Fluorescence was measured using Perkin Elmer Envision for 5 s after which a I- buffer (137 mM NaI, 2.7 mM KI, 1.7 mM KH2PO4, 10.1 mM Na2HPO4, 5 mM D-glucose) was injected into the well and fluorescence monitored for another 7 s. YFP quenching was determined at the end of the interval as F/F0, and CFTR function as 1-(F/F0).
C4. Figures 1D, 3, 5: Further details would need to be provided within the figure legends on what the data points (ie squares, circles) indicate.
R4. The legends of Figure 1D, 3, and 5 were adapted in response to the comments from both reviewers. See Reviewer 1 – R4 for the updated figure legend of Figure 1.
Updated Figure 3 legend:
Figure 3. Rescue in CFTR function of rare CFTR mutants by VX-809 and/or VX-770. (A-C) HEK293T cells stably overexpressing untagged CFTR variants were treated with 2.5 µM VX-809 for 24 h and/or 3 µM VX-770 for 20 min prior to determining CFTR function by HS YFP quenching after stimulation with 10 µM forskolin (Fsk). Graphs depict the mean ± S.D. of three independent experiments represented as dots (DMSO) or squares (compounds). Response to treatment over DMSO was compared for each mutant by multiple t-testing. *p<0.05, ** p<0.01, ***p<0.001, ****p<0.0001.
Updated Figure 5 legend:
Figure 5. Quantification of plasma membrane density and traffic efficiency by flow cytometry of CFTR mutants. (A) 3HA-F508del-CFTR HEK293T cells were treated with increasing amounts of corrector VX-809 (range 5 nM – 50 µM) or DMSO for 24 h and analyzed by flow cytometry. Left: Representative histograms of different VX-809 concentrations and their corresponding PM density levels is given. Right: EC50 determination of the dose-response to VX-809. Mean ± S.D. of n=3 independent experiments. (B) Representative histograms of the PM density of the different 3HA-tagged CFTR variants under basal (DMSO) conditions or treated with VX-809. (C, D) PM density (C) and total CFTR (D) at baseline (DMSO) and after VX-809 treatment (2.5 µM, 24 h) relative to WT-CFTR DMSO. Independent experiments are shown as dots (DMSO) or squares (VX-809). (E) Fold increase in PM density after VX-809 treatment over DMSO control. (F) This graph represents the increase in traffic efficiency (TE) by VX-809 compared to DMSO (ΔTE = TEVX-809 – TEDMSO), with TE = % PM/ % Total CFTR (with WT DMSO = 1). Graphs depict the mean ± S.D. of three independent experiments. VX-809 rescue over DMSO was compared for each mutant by multiple t-testing. *p<0.05, ** p<0.01, ***p<0.001, ****p<0.0001. Abbreviations: PM: plasma membrane, EC50: half-maximal effective concentration, NT: non-transduced.
C5. Line 220: Figure citation needs to be kept consistent, ie Figure 2A-C before Figure S4B
R5. We agree with the reviewer and have changed this throughout the manuscript.
